# Successful Intratracheal Treatment of Phage and Antibiotic Combination Therapy of a Multi-Drug Resistant *Pseudomonas aeruginosa* Murine Model

**DOI:** 10.3390/antibiotics10080946

**Published:** 2021-08-05

**Authors:** Christopher Duplessis, Jonathan M. Warawa, Matthew B. Lawrenz, Matthew Henry, Biswajit Biswas

**Affiliations:** 1Naval Medical Research Center, 503 Robert Grant Avenue, Silver Spring, MD 20910, USA; matthew.s.henry23.ctr@mail.mil (M.H.); Biswajit.biswas.civ@mail.mil.civ.mil (B.B.); 2Department of Microbiology and Immunology, Center for Predictive Medicine for Biodefense and Emerging Infectious Disease, University of Louisville School of Medicine, Louisville, KY 40202, USA; jmwara01@louisville.edu (J.M.W.); mblawr02@louisville.edu (M.B.L.); 3The Geneva Foundation, Tacoma, WA 98402, USA

**Keywords:** bacteriophage, *Pseudomonas aeruginosa*, pneumonia, multi-drug resistance, phage antibiotic synergy, intubation-mediated, intratracheal

## Abstract

Background: Pseudomonas aeruginosa (PsA) is a common etiology of bacteria-mediated lower respiratory tract infections, including pneumonia, hospital acquired pneumonia (HAP), and ventilator-associated pneumonia (VAP). Given the paucity of novel antibiotics in our foreseeable pipeline, developing novel non-antibiotic antimicrobial therapies saliently targeting drug resistant PsA isolates remains a priority. Lytic bacteriophages (or phages) have come under scrutiny as a potential antimicrobial for refractory bacterial infections. We evaluated intratracheally and intraperitoneally (IP) administered phage therapy (with/without meropenem) in an acute immunocompromised mouse model of multi-drug resistant (MDR) PsA pulmonary infection. The MDR *P. aeruginosa* respiratory disease model used in these studies was developed to investigate novel therapies that might have efficacy as either monotherapies or as combination therapy with meropenem. Methods: We utilized eight-week-old, 18 g BALB/cJ female mice and an MDR strain of *PsA* (UNC-D). Mice were immunosuppressed with cyclophosphamide. We employed a three-phage cocktail targeting PsA (PaAH2ΦP (103), PaBAP5Φ2 (130), and PaΦ (134)), confirmed to exhibit in vitro suppression of the infecting isolate out to 45 h. Suppression was confirmed with phages acting in isolation and in combination with meropenem. Results: IP administration of phage did not protect mice from death. A one-time delivery of phage directly to the lungs via a single intubation-mediated, intratracheal (IMIT) instillation protected mice from lethal infection. Protection was observed despite delaying therapy out to 6 h. Finally, we observed that, by slowing the progression of infection by treatment with a sub-efficacious dose of meropenem, we could protect the mice from lethal infection via IP phage administration coupled to meropenem, observing partial additive effects of phage–antibiotic combination therapy. Conclusions: A personalized phage cocktail administered via IMIT exhibits high therapeutic efficacy, despite delayed treatment of 6 h in a lethal MDR PsA pneumonia model. IP phage alone did not forestall mortality, but exhibited efficacy when combined with meropenem and IMIT-administered phage. These additive effects of combined IP phage and meropenem confirm that phage may indeed reach the lung bed via the systemic circulation and protect mice if the infection is not too acute. Therefore, adjunctive phage therapy with concerted attention to identifying optimal phage targeting of the infecting isolate in vitro may exhibit transformative potential for combating the specter of MDR bacterial infections. Phage should serve as an integral component of a four-pronged approach coupled with antibiotics, source control, and immune optimization.

## 1. Introduction

The advent and increasing prevalence of antimicrobial resistance commensurate with the absence of novel antibiotics on the horizon raises the specter of untreatable infections. We must now grapple with infections stemming from extensively multi- and pan-drug resistant bacterial strains. Respiratory infections are an important source of multi-drug resistant (MDR) infections, leading to morbidity and mortality [1]. MDR Pseudomonas aeruginosa (PsA) is a common etiology of bacteria-mediated lower respiratory tract infections, including pneumonia, hospital acquired pneumonia (HAP), and ventilator-associated pneumonia (VAP) [2,3]. Given the paucity of novel antibiotics in our foreseeable pipeline, developing novel non-antibiotic antimicrobial therapies saliently targeting drug-resistant PsA isolates remains a priority.

Potential antimicrobials of import include lytic phage and phage lysins. Lytic phages are immediately virulent to the host, whereby the phage nucleic acids do not integrate within the bacterial genome, rather they immediately commandeer the metabolic machinery of the bacteria to synthesize the phage-specific protein and nucleic acids. This process immediately shuts down the bacterial replicative machinery and stops bacterial growth. As a class of antimicrobial agent, bacteriophages offer several potential advantages, including: (1) the absence of safety concerns as delineated in a recent review [4]; (2) bactericidal activity irrespective of antibiotic resistance profiles; (3) localized concentration increase at the site of infection due to phage replication; (4) minimal collateral damage to the healthy microbiome (specificity of phage infection may sculpt the microbiome in a targeted manner to allow the overgrowth of less pathogenic bacteria in the ecological space emptied of pathogenic bacteria) [5]; (5) potential synergy or additive effects with antibiotics; (6) potential reversion of bacterial susceptibility to antibiotics; (7) activity against bacterial biofilms; (8) intrinsic adjuvant activity of phage, including: (i) activating innate immunity via Toll-like receptors (TLRs) and (ii) activation of adaptive immunity via exposing lysed bacterial constituents attached with phage to antigen-presenting cells (APCs); and (9) anticipated cost effectiveness of pharmaceutical development [6,7,8,9,10,11,12].

Adjunctive phage therapy is slowly filtering into the antimicrobial armamentarium for the treatment of refractory infections, as well as saliently recalcitrant lower respiratory infections attributed to MDR and XDR PsA in pre-clinical models [13,14,15,16] and clinical therapy [1,2,3,5]. We postulate that the optimal utilization of phage therapy in clinical practice likely requires phage cocktails (polyvalent mixtures of phages) confirmed to target the infecting isolate in vitro combined with antibiotics. The phage mixtures combined with antibiotics should optimally and reciprocally suppress emerging bacterial resistance evolving during therapy, and target multiple strains resident within a genetically and phenotypically heterogeneous population. The latter assertion is particularly salient in biofilm-mediated infections [1]. Finally, the utilization of phage and antibiotic combinations may be particularly important in an immune-suppressed host.

During our investigation, phage therapy in both pre-clinical and clinical applications for the treatment of respiratory infections has been heterogeneous in design. Specifically, we note heterogeneity in phage administration regarding the route (intravenous (IV), intratracheal, both), frequency of administration, and duration [1,2,3,5,13,14,15]. We therefore wished to systematically evaluate the efficacy of phage therapy in a lethal murine PsA pneumonia model via intraperitoneal (IP) (which is similar to IV delivery), intratracheal, and combinatorial routes of administration. Additionally, we assessed the efficacy of localized instillation of phage for the first time via direct administration to the lungs via intubation-mediated, intratracheal (IMIT) delivery. Finally, we delayed the IMIT-mediated phage delivery relative to bacterial inoculation by 6 h, the longest delay to our knowledge in this model of lethal PsA pneumonia to anticipate any potential survivability. Therefore, the overarching objectives of this effort was to assess the therapeutic efficacy of phage with and without meropenem in a lethal cyclophosphamide (Cy) immunocompromised mouse model of MDR PsA respiratory disease (pneumonia). Specifically, we evaluated the therapeutic efficacy (survivability and time to mortality) of phage treatment administered via IP injection, IMIT instillation, IMIT and IP combination adminstration, and as adjunctive treatment with meropenem in a model achieving persistently high lethality during meropenem treatment only (given PsA resistance to meropenem). Within all studies undertaken, we evaluated overall host survival, bacterial burden in the lungs and spleen, and the lung pathology via standard histopathological evaluation.

## 2. Results

### 2.1. Identifying Personalized Phage Cocktails and Antibiotic-Phage Activity

The phages were isolated from local wastewater sources using the infecting strain as the target bacteria. Similar to our prior efforts, we executed screening against a diverse collection of clinical isolates by efficiency of plating (EOP) assays. All phages were maintained in their respective hosts in TS media and were subject to a minimum of three plaque purification steps to confidently identify a single strain of phage. We identified three phages with differential potency in in vitro killing of the infecting bacterial isolate (UNC-D). All three phages were characterized and confirmed to target the infecting strain, achieving robust killing activity in vitro prior to each experiment via the Host Range Quick Test (HRQT) [17] and absence of any evidence of antagonism (Figure 1). The novel phages were sequenced, the results of which indicated that all phages belonged to the Myoviridae family of phages. Specifically, 10^4^ colony-forming units (cfu) of PsA UNC-D were inoculated separately with phage PaAH2ΦP (103), PsBAP5Φ2 (130), and PAΦ134 (at a multiplicity of infection (MOI) of 100 for each phage) in the presence and absence of meropenem (5 µg/mL).

PsA UNC-D was sensitive to all three phages. Phage PsBAP5Φ2 (130) inhibited the growth of bacteria up to 22 h before phage-resistant bacteria started dominating the culture (Figure 1, yellow line,). Otherwise, the other two phages independently, the three-phage cocktail, and each phage combined with meropenem effectively suppressed the bacterial growth (Figure 1). The assay also confirms the resistance of the UNC-D isolate to meropenem (MIC previously shown to be 8 µg/mL [18]. Therefore, we established the in vitro proof of efficacy of personalized phages targeting (killing) the infecting isolate, and in vitro additive efficacy in killing the targeted isolate by exploiting the combination of PsBAP5Φ2 (130) with meropenem.

### 2.2. Confirming the Lethality of the Model

To confirm the lethality in this model with the meropenem-resistant PsA isolate, we assessed different concentrations of meropenem in our murine model. The bacterial inoculum was confirmed by serial dilution and colony enumeration on TSA plates prior to each trial. While a subset of animals was not sacrificed to determine the number of bacteria in the lungs post-infection, we have previously shown that about 98% of the inoculum is reproducibly delivered to the lungs using the IMIT procedure [19]. All studies included a mock-treated group to ensure that lethal infection was established, and that infection kinetics remained unchanged. We observed no difference in the mock-treated animals between studies. We specifically assessed three concentrations of meropenem (1000 mg/kg/day, 1250 mg/kg/day, and 1500 mg/kg/day), all administered q8 × 5 days subcutaneously and initiated 3 h post-bacterial inoculation (via IMIT). We observed near uniform lethality with all concentrations investigated (Figure 2). Therefore, we utilized a meropenem dose of 1250 mg in all subsequent investigations. As to be addressed in the discussion, the additive effect of phage and antibiotic therapy is likely best observed at lower antibiotic concentrations, supporting the selection of the meropenem dose.

### 2.3. Systematic Investigations into Phage Efficacy

Upon confirming the lethality of our PsA pneumonia model and designating a dose of meropenem, we pursued four separate investigations to assess phage efficacy.

Trial (1): First, we assessed the potential therapeutic efficacy in the IP administration of phage (1 × 10^9^ plaque-forming units (pfu)) q8 for 120 h, delayed relative to bacterial inoculation by 3 h. Figure 3 shows that IP administration of phage did not protect mice from death. In one of two replicate experiments, treatment with the phage cocktail administered IP alone provided a small delay in the mean time to death that approached significance (*p* = 0.0709), accompanied by a significant decrease in bacterial burden in the lungs and spleen. However, a similar phenotype was not observed in the second replicate experiment.

Trial (2): Since IP administration alone did not protect against mortality, we next tested whether a one-time delivery of phage directly to the lungs via IMIT (2.5 × 10^9^ pfu) delayed by 3 h from bacterial inoculation followed by IP injections of phage (1 × 10^9^ pfu) q8 for 120 h could protect mice from lethal infection. (Figure 4). Here, we observed near 100% survival with treatment in replicate experiments with significant reductions in bacterial burdens in the lungs and spleens and reduced pulmonary pathologic scores for all treatment groups.

Trial (3): Since combination delivery protected mice from lethal infection, we next tested if a one-time administration of phage directly to the lungs (via IMIT at 2.5 × 10^9^ pfu) was sufficient to protect mice from lethal infection (Figure 5). Similar to the results observed utilizing the combination of phage administered via IMIT + IP, we observed that all mice receiving phage via IMIT survived the infection (Figure 5A, 3 h). Moreover, IMIT delivered phage also provided protection if pahge administration was delayed to 6 h after bacterial administration (Figure 5A, 6 h). Directly corelating with survival, we observe significant reductions in the pulmonary bacterial burden and pathologic scores for IMIT phage delivery relative to the saline placebo (“Mock”).

Trial (4): Finally, we assessed whether slowing the progression of infection by treatment with a sub-efficacious dose of meropenem, could improve the efficacy of IP administration of phage. Towards this end, meropenem was administered subcutaneously (1250 mg/kg/day q8 for 120 h) and phage were administered by IP injection (1 × 10^9^ pfu administered q8 for 120 h), both delayed by 3 h relative to bacterial inoculation (Figure 6). The combinatorial treatment of meropenem and phage significantly enhanced therapeutic protection (achieving a survival benefit of >50% survival) against pulmonary infection *p* = 0.0022 (log-rank test). The phage–meropenem combination also achieved a significant reduction in bacterial burden (relative to placebo) in the lungs and spleen, accompanied by a reduced pathology score. These data support that phage administered IP can penetrate the pulmonary tissues, and in combination with a sub-efficacious dose of antibiotic that can slow bacterial proliferation, but not protect against lethal outcome, phage can provide a significant survival benefit over antibiotic only.

## 3. Discussion

The MDR *P. aeruginosa* respiratory disease model used in these studies was developed to investigate novel therapies that might have efficacy as either monotherapies or as combination therapy with meropenem [18]. The bacterial strain used, UNC-D, is a clinical isolate with resistance to clinically important carbapenem antibiotics, such as meropenem and imipenem. We have examined numerous novel therapeutics in this model with the strategy of examining combination therapies at a concentration of meropenem that provides a near-failure dosing. We have empirically determined that this provides the most opportune testing environment for new combination therapies. Furthermore, the advent of increasing drug resistance amongst clinically relevant bacterial pathogens means that current antibiotic modality therapeutics are starting to lose efficacy as monotherapy, and would benefit from formulation with an adjunct. Indeed, the preponderance of new antibiotic therapies being brought to market are not new antibiotics per se, but rather combination therapies designed to improve the efficacy of a failing antibiotic. Examples of these new combination therapies include Recarbrio (imipenem with the β-lactam inhibitor relebactam) and Vabomere (meropenem with the β-lactam inhibitor vaborbactam). Thus, our meropenem combination therapy models the current methodologies used by FDA approved formulations to improve classic antibiotic therapies that are beginning to fail as monotherapies due to the advent of MDR pathogens [18].

We confirmed that our model was robust and reliable in assessing delays in mortality, in this case by assessing phage therapy and survivability with harmonized results from survivability with evaluations of organ-specific bacterial burdens and lung pathologic scores. The phage treatments were confirmed safe without any adverse events identified attributed to phage administrations. Phage therapy has been consistently and reliably immunologically and clinically well-tolerated in pre-clinical and clinical applications, lacking evidence of immune interference either primarily or coupled with antibiotic treatment.

We systematically pursued four separate investigations (dictated in part by the results of antecedent investigations) to assess phage efficacy in this effort to address the heterogeneity in the existing literature (pre-clinical and clinical) regarding the potential efficacy of adjunctive phage therapy in clinical practice. We specifically assessed (1) the efficacy of phage therapy administered IP q8 for 120 h; (2) the efficacy of combining IP phage therapy (q8 for 120 h) with a single IMIT dose; (3) the independent efficacy of a single IMIT dose of phage; and (4) efficacy with combinations of IP phage (q8 for 120 h) coupled to meropenem. The motivation for trial four, assessing combinatorial phage and antibiotic, emanates from the fact that the frequency of phage infection decreases with decrements in the bacterial population; therefore, a dynamic equilibrium may be reached between phage and their bacterial host without achieving complete eradication. This supports the contention that optimal treatment likely requires combinations of phage and antibiotics (coupled with optimization in host immunity and source control) [14].

Our results reveal that a one-time administration of phage via IMIT delivery was 100% efficacious in preventing mortality in an aggressive lethal murine model. IP phage administration coupled to IMIT administration may provide clinical improvement when assessing the bacterial burdens and pulmonary pathologic scores. Finally, IP phage administration was additive to antibiotic therapy in preventing mortality while reducing bacterial burdens and pulmonary pathologic scores confirming a viable clinical therapeutic for PsA-mediated pneumonia. This investigation confirms the utility of our aggressive murine model, the utility of IMIT as a reliable precise delivery method for local intratracheal phage delivery, the efficacy of intratracheal phage delivery, and the clinical applicability of combining phage and antibiotics for treatment of refractory bacterial infections.

To date, the literature in phage therapy has been heterogeneous regarding the optimal routes of delivery for varied clinical syndromes (saliently pneumonia). We haven’t achieved a consensus regarding the ideal route for phage administration in pneumonia (whether IV or intratracheal delivery in isolation or its combination) [3,5]. Establishing efficacy in IP phage administration assumes prominence as this delivery would be clinically germane, and the viable route in most patients who aren’t intubated. Prior efforts suggest that phage administered via any route expediently circulates to reach all compartments, and IV phage delivery achieves similar alveolar concentrations in the lungs as via direct intratracheal instillation [5,20]. We acknowledge prior efficacy with intratracheal phage therapy in murine pneumonia models [13,14,15,16,21,22,23,24,25], prompting trials 2 and 3, wherein we assessed efficacy with a significant delay (6 h) in therapy in an aggressive and lethal murine model, wherein such a delay is anticipated to engender significant mortality. Our model is validated to be lethal absent punctual therapeutic intervention. Any therapeutic intervention including antibiotics would not be anticipated to achieve survivability if delayed beyond 6 h [18,22].

In trial (1), treatment with the phage cocktail administered IP alone delayed at 3 h provided a small delay in the mean time to death that approached significance (*p* = 0.0709), but did not forestall mortality. We administered the IP phage (10^9^ pfu q8) given the known rapid clearance from the circulation over the ensuing hours [20] coinciding with meropenem administration. We evaluated five days of therapy, as prior investigations suggested a requisite protracted administration duration to achieve efficacy when utilizing phage therapy via the systemic route (i.e., IV or IP) [1,2,3,5].

In trial (2), we assessed if combining IP and intratracheal phage therapy via a single IMIT administration (as may be anticipated with severe clinical infections in intubated patients) could provide efficacy regarding delays in mortality. We observed significant survival (nearly 100% in all groups observed) with the addition of a single IMIT dose of phage delivery. This prompted trial (3) to assess the efficacy of a single IMIT-mediated dose of phage administration in isolation. We observed universal survival underscoring the efficacy in local phage delivery. Excellent survival was observed, despite a protracted delay in therapy (relative to bacterial inoculation) out to 6 h. Of note, IP administered phage may afford clinical efficacy in combination with IMIT instillation of phage, evidenced by reduced bacterial burdens and pathologic scores suggesting that no distinct antagonism occurred when combining IMIT instillation of phage with IP administered phage.

Intuitively, the topical (local) delivery of the therapeutic to the site of infection may engender superior clinical outcomes. Aerosolized phage delivery (via nebulization) has been successfully delivered in humans [5,22], and was superior to systemic (IP) delivery in treating a *Burkholderia cenocepacia* murine pneumonia model [16]. In the treatment of pneumonia, intratracheal delivery and direct deposition, as well as concentration within the lungs at the source of infection via the aerosolization (nebulization) of therapeutics (including antibiotics and phage), may circumvent potential issues with tissue penetration [2], while possibly circumventing host immunity (pre-formed antibodies targeting phages) [2]. Additionally, intratracheal phage delivery may circumvent impediments in pulmonary access engendered systemically due to cavitation, necrosis, and excessive inflammation. Intranasal phage delivery has been successful, however, it suffers from unreliable (precise) pulmonary deposition given uncontrolled mucocilliary clearance and upper airways clearance [2,23]. Various nebulization strategies engender phage loss and inactivation [21]. The IMIT delivery platform mimics clinical scenarios in patients intubated providing for reliable, precise, and reproducible establishment of pulmonary infection and subsequent therapeutic administration. Therefore, a major objective in this effort mirroring prior investigations establishing efficacy of intratracheal phage delivery for pulmonary infections was to assess superiority of intratracheal phage delivery via a reliable delivery method which may allow delayed administration out to 6 h. A 6 h delay has been shown to reduce intratracheal phage efficacy dramatically [15,16]. Many of the pre-clinical murine pneumonia (secondary to PsA) models have investigated the efficacy of intratracheal phage delivery at 2 h [14,15,16,23,26] or reduced survivability out to 4 h [13].

We were limited to a single IMIT dose of phage given the stressors associated with its administration, but prior investigations suggest protracted pulmonary residency in uninfected mice [14]. A single localized intrapulmonary delivery of phage (10^9^ pfu) in healthy mice could prophylax against infection for 4 days with PK assessment confirming a mere 0.5 log decrement per 24 h [14]. In the context of infection, the known exponential phage increase is anticipated, particularly if married to antibiotics and host immune optimization. Single intratracheal phage administration has successfully improved mortality in murine PsA mediated pneumonia models [13,15,16]. Additionally, our one-time IMIT administration conforms to recent literature seeking efficacy via similar dosing frequency in other pathogens including *Acinetobacter baumanii* (AB) [26,27,28].

Clinical utilization of phage administered intratracheally requires confirmation of efficacy targeting multiple pathogens. In addition to PsA, there’s a growing literature evaluating phage efficacy in AB mediated pneumonia. Phage administration 30 min post AB infection protected against mortality in a murine model [26]. In this investigation, a single intratracheal dose of phage cocktail targeting the AB isolate proved efficacious in reducing the pulmonary bacterial burden. A clinical case study exhibited efficacy in multiple (16 days) intratracheal dosing of targeted phage against an AB mediated pneumonia [28].

Clinical applications utilizing intratracheal phage administration for lower respiratory infections, whether administered in isolation or when combined with intravenous (IV) phage delivery, have exploited protracted durations of administration to achieve a favorable clinical result [1,3,5]. Therefore, our results provide a foundation for clinical studies to assess adjunctive phage therapy with emphasis upon intratracheal delivery, confirmation of in vitro targeted killing of a phage cocktail, and confirmation of an absence of antagonism utilizing the combinations of phage and antibiotics.

Finally, we assessed for improvement in efficacy with combination of IP administered phage and meropenem. We may not proffer “additive effects or synergy” given the lack of efficacy observed in either treatment in isolation. IP administered phage combined with a sub-efficacious dose of meropenem delayed mortality, and improved survival rates, while exhibiting reduced bacterial burdens and pathologic scores thus confirming certainly no antagonism. We posit that clinically relevant introduction of phage in therapeutic algorithms for recalcitrant MDR infections will require optimization of the additive or synergistic effects of in vivo phage–antibiotic therapies, most saliently in immune-suppressed patients [21,29,30,31,32,33,34]. Additive or synergistic effects of phage–antibiotic combination therapies (1) may be exploited utilizing lower antibiotic concentrations engendering lower side effects, (2) reciprocally suppress bacterial resistance emanating from evolutionary and genetic trade-offs and fitness costs [5,29,30], and (3) promote re-sensitization to antibiotics regardless of in vitro resistance [29,30]. Sub-inhibitory antibiotic concentrations may circumvent deleterious stressor responses interfering with concomitant phage activity [32].

When logistically feasible, sequential administration of phage preceding antibiotics at initiation may be ideal allowing phage access to commandeering of the metabolic machinery of the bacteria otherwise blunted in presence of antibiotics, and optimizing discrete phage and antibiotic–host interactions [9,29,32,33,34].

We note that employing the HRQT to assess bacterial suppression in vitro may be imperfect (i.e., failing to mimic the in vivo environment, bacterial phenotypic expression, and host-specific immune status), yet, in our experience, as well as those of others, in vitro confirmation of bacterial suppression translates to in vivo efficacy [17].

Technical limitations prevented pursuit of phage PK assessments. Continuous phage activity upon both live and dead bacteria undermines the faithful representation of phage titers, requiring meticulous specimen processing to inactivate phage–bacterial interactions without engendering phage degradation, employing ex vivo freezing algorithms with glycerol or other suitable preservative. More saliently, future investigations will pursue post-treatment sampling to characterize evolution of phage and antibiotic resistance and/or sensitivity, most prominently in context of immune-suppression and accelerated resistance evolution without the accompanying immunophage (“neutrophil–phage”) synergy [14].

## 4. Materials and Methods

### 4.1. Experimental Design

The lethality in this model with the PsA isolate resistant to meropenem was confirmed by assessing different concentrations of meropenem, specifically assessing meropenem dosing of 1000 mg, 1250 mg, and 1500 mg/kg/day. The intent was the selection of the lowest dose with reproducible mortality for subsequent investigations for combinations with phage. As addressed in the discussion, phage antibiotic combinatorial efficacy is likely best observed at lower antibiotic concentrations.

Four separate trials were pursued. In all subsequent trials, BALB/cJ-Cy mice (n = 8 per group) were inoculated with 10^5.5^ cfu of UNC-D by IMIT. All phage adminstrations were delayed from bacterial inoculation by 3 or 6 h, as specified in the narrative. IP administrated phages were administered at 1 × 10^9^ pfu, q8 for 120 h. IMIT administrated phages were delivered at 2.5 × 10^9^ pfu. In all trials, the bacteria burden was assessed from pulmonary and splenic harvesting, with pulmonary histopathologic scores determined. Finally, monitoring continued for 168 h (seven days).

#### 4.1.1. Trial (1)

In the first trial, the potential therapeutic efficacy of the IP administration of phage in isolation was assessed, administered with a delay of 3 h from bacterial inoculation.

#### 4.1.2. Trial (2)

Since IP administration alone did not protect against mortality, a one-time delivery of phage directly to the lungs via IMIT, followed by IP injections of phage q8 for 120 h was assessed for protecting mice from lethal infection.

#### 4.1.3. Trial (3)

Since combination delivery protected mice from lethal infection, a one-time administration of phage directly to the lungs (via IMIT) delayed by 3 or 6 h was then assessed for efficacy in protecting mice from lethal infection.

#### 4.1.4. Trial (4)

Finally, a sub-efficacious dose of meropenem was administered in an effort to slow progression of infection, and protective efficacy was assessed upon administering IP phage and meropenem (1250 mg/kg/day q8 for 120 h).

### 4.2. Mice and Induction of Leukopenia

All animal care was conducted in compliance with all established IACUC policies at the University of Louisville (UofL). UofL is fully accredited by the Association for Assessment and Accreditation of Laboratory Animal Care International (AAALAC). All studies were conducted in accordance with IACUC protocol #16592, the Standard Operating Procedures (SOPs) of the Center for Predictive Medicine, and the applicable regulatory requirements. Seven-week-old BALB/cJ female mice were supplied by Jackson Laboratories. Mice were allowed to acclimate to the facilities for three days prior to implantation with a temperature transponder (BioMedic Data Systems, Seaford, DE, USA) utilized for digital identification and core body temperature monitoring. Mice were group housed (up to four/cage) in solid–bottom polycarbonate individually ventilated micro-isolation cages (Tecniplast, West Chester, PA, USA). To establish leukopenia, the mice were treated with cyclophosphamide monohydrate (Sigma, St. Louis, MO, USA) administered via IP injection (0.1 mL (150 mg/kg) on days five and three prior to the bacterial challenge (in all studies referred to as occurring on day 0). Due to the systemic effect of cyclophosphamide treatment, supportive care was provided to minimize dehydration, including 1 mL of sterile saline delivered via subcutaneous injection on days three through one prior to infection. Leukopenia was confirmed by complete blood cell counts (CBC) on day one prior to infection.

### 4.3. Bacteria

*P. aeruginosa* UNC-D is a sputum isolate from a patient with cystic fibrosis kindly provided by Dr. Peter Gilligan at the University of North Carolina [18]. Minimum inhibitory concentrations of the UNC-D strain are: ceftazidime (32 µg/mL), meropenem (8 µg/mL), imipenem (16 µg/mL), tobramycin (32 µg/mL), piperacillin (16 µg/mL), aztreonam (4 µg/mL), colistin (1 µg/mL), and fosfomycin (256 µg/mL). For infection, bacteria were cultured in Lennox broth overnight at 37 °C with shaking and washed into sterile phosphate buffered saline, enumerated by OD600 absorbance, and diluted to an appropriate final concentration in 50 ul for injection. The inocula were confirmed by serial dilution and colony enumeration on TSA plates prior to each trial.

### 4.4. Antibiotic and Phage Preparations

Meropenem for injection, USP (NDC 0409-3506-01) was supplied by Hospira (Lot Number: 609H012). Meropenem was prepared on the day of administration by diluting to appropriate concentrations in sterile saline (Henry Schein) for injection.

The final phage preparation was comprised of the equimolar mixtures of the following phage (PaAH2ΦP (103), PaBAP5Φ2 (130), and PaΦ134) (Lot # BDRD201805) manufactured and shipped one week prior to ordering the mice, ready for administration. The phage cocktails were provided at 5 × 10^9^ pfu/mL for IP, and 5 × 10^10^ pfu/mL for IMIT administration, respectively, stored at 4 °C until use (stable for one year). The endotoxin levels were 39,000 EU/mL after tangential filtration and dialysis.

Briefly, all the above-described phages were derived from the Naval Medical Research Center Biological Defense Research Directorate (BDRD) current library of phages collected from various environmental sources, which allow the open-ended development of precision phage cocktails. The bacterial strain is stored as a streak TS agar plate at 4 °C for short-term work or in TS broth with 20% *v*/*v* glycerol at −80 °C for long-term storage. The strain was grown in 3 mL subcultures of TS broth, which were incubated at 37 °C with shaking at 200 RPM to an OD_600_ of approximately 0.5 before use. The growth rate (bacterial cellular respiration) analysis was executed via the OmniLog microplate assay [17], with subtle nuances leading to the automated colorimetric (tetrazolium dye) assay we call the HRQT. This testing platform incorporates a colorimetric assay and monitors the growth of bacteria in vitro with/without interventions, including antibiotics and bacteriophages. Bacterial respiration reduced the tetrazolium dye, which changed the color of the media to purple recorded by a camera and depicted as relative units of bacterial growth. The HRQT assay is run in 96-well plates that are robotically loaded with 10^4^ bacterial cells per well, tetrazolium dye, varying concentrations of phage, and other interventions (including antibiotics). Plates are then read every 15 min for several hours in an OmniLog instrument (Biolog Inc., Hayward, CA, USA) and “kill curves” are generated (see Figure 1). The *y*-axis reveals the relative respiration units and the *x*-axis displays time.

Specifically, in these experiments, the phages were inoculated into wells at a multiplicity of infection (MOI) of 100. Each well in the 96-well BioLog plates was adjusted to a total volume of 100 µL of TS broth (bacterial growth medium) with the 1% BioLog dye D tetrazolium-based dye. Once BioLog plates were inoculated, they were incubated in the OmniLog system at 37 °C for 45 h. Colorimetric data points were collected every 15 min over the time course.

Bacterial growth suppression is indicated by the isolated bacterial curve remaining flat for an extended period of time for a minimum [although somewhat arbitrarily defined but based on our experience with clinical cases including emergency INDs (eINDs)] of 4 h. Efficacious phage(s) suppress bacterial proliferation for a sufficient period reflecting in vitro (and in our experience by extension in vivo) efficacy. The speed of this strategy, and its ability to identify effective phage combinations, and phage–antibioticcombinations affording efficacious bacterial killing, enables practical clinically viable personalized phage therapy [17].

Specifically, the PsA isolate (approximately 10^4^ cfu *P. Aeruginosa* A52-WS 114) was inoculated with the bacteriophage individually, and in combination with/without meropenem (5 µg/mL) in 96-well microtiter plates containing tryptic soy media in 1% (*vol*/*vol*) tetrazolium dye and incubated at 37 °C in a Biolog machine for 45 h to evaluate the phage-bacterial interaction to assay the phage mediated killing of bacteria. [17].

All three phages were characterized and confirmed to target the infecting strain achieving robust killing activity in vitro prior to each experiment. Any permutation of phage valiancy may have exhibited efficacy (i.e., one or two phage cocktails) based upon our in vitro results. This three-phage cocktail was selected to overcome any kind of bacterial resistance barrier against phage during in vivo phage treatment, while optimizing assessments for therapeutic efficacy via alternative routes of delivery (IP versus intratracheal), with/without meropenem. Additionally, we wished to evaluate potential vagaries in respect of additive, synergy or antagonism (interference) in the efficacy of multivalent phage cocktails with the antibiotic meropenem (to which the bacteria were resistant) reflecting real world clinical scenarios. Finally, we wished to maximize pressure for potential reversion to antibiotic sensitivity during treatment, a known phenomenon in phage therapy likely more achievable employing a higher phage valiancy. Employing the HRQT in real-time mirrors the real-life clinical scenario, whereby we may expeditiously assess phenotypic functional bacterial suppression, a prominent advantage of the personalized approach, without meticulously characterizing bacterial targets (as pursued when employing fixed phages). We acknowledge that these targets may not be reliably phenotypically expressed in vivo. Additionally, this personalized approach potentiates sequential identification of novel efficacious phage cocktails accommodating resistance evolution during phage therapy.

### 4.5. Administration of Bacterial Inoculum and Therapeutics

Bacteria (10^5.5^ cfu) were administered to animals in 50 uL of saline via IMIT instillation as previously described [19]. Meropenem was delivered subcutaneously commencing at 3 h post challenge q8 at alternating sites (hind right quadrant, hind left quadrant, and scruff of the neck) for five days. For systemic phage delivery, the phages were administered via IP injection, q8 (0.2 mL = 10^9^ pfu) to coincide with Meropenem delivery (initiated 3 h post challenge). For local intratracheal delivery, the phage (50 µL = 2.5 × 10^9^ pfu) was administered via the established IMIT instillation method [19].

### 4.6. Clinical Observations, Bacterial Enumeration, and Histopathology

Animals were monitored for the development of illness q8 after infection until the end of study. Mice exhibiting temperatures < 26.7 °C or a pre-defined clinical endpoint including loss of righting reflex were considered moribund and were humanely euthanized by carbon dioxide asphyxiation and exsanguination. Euthanized animals were scored as succumbing to infection at the next health check. Survivor mice were euthanized on day seven. Tissues were collected at euthanasia for bacterial counts and histopathology.

Necropsy was executed via carbon dioxide asphyxia upon which maximum blood draws were acquired via cardiac puncture and placed into K_2_EDTA tubes, and lung and splenetic organs harvested (collected into pre-weighed 1 oz. Whirl-Pak bags). The lungs were collected in the pre-weighed 1 oz. Whirl-Pak bags and weight recorded to collect the total lung weight. The lungs were retrieved from the Whirl-Pak bag and placed on a sterile dissection board to have a representative section removed for histopathology (a nickel-thick central section from both lung halves were excised and fixed in 10% neutral-buffered formalin (NBF)). The lungs were returned to the Whirl-Pak bags and a second weight recorded to measure lung weight minus the excised section for pathology. The spleens were removed to a Whirl-Pak bag and weighed, however no pathology sections were required for the spleen. The lung sections collected for pathology were fixed for 24 h in 10% neutral-buffered formalin (NBF) at room temperature followed by transfer into labeled tissue cassettes and storage in 70% ethanol before submission for H&E staining. The lung and spleen samples in Whirl-Pak bags were homogenized after the addition of 1 mL of PBS. The samples were treated with Triton X-100 (1% final concentration) and serially diluted in round bottom 96 well plates to conduct bacterial enumeration analysis. All murine studies were performed at the Regional Biocontainment Laboratory at the University of Louisville. In all studies, the cages housing the separate groups (8-mice) divided into 4-mice per cage not receiving phage, and all cages housing mice receiving placebo (3-mice) were handled first prior to the groups (cages) receiving phage to avoid any cross-contamination.

Lung histopathology was scored in a blinded fashion by a board-certified veterinary pathologist. A four-point, four-criteria system (inflammation; infiltrate; necrosis; and other, including hemorrhage) with a maximum score of 16 points was used to evaluate lung pathology. Points for each criterion were assigned based as no (0), minimal (1), mild (2), moderate (3), and severe (4) pathologic findings.

### 4.7. Statistical Analyses

No formal statistical power analyses for group sizes were performed as we intended to utilize the minimal number of mice (8) from which to extract qualitative evidence of efficacy, reconciling the desire to assess permutations of therapeutic regimens. Statistical comparisons for survival times were calculated by the log-rank test. Bacterial burdens (lung and spleen), log transformed, and pathologic scores were assessed between groups via the unpaired Student’s *t* test or one-way ANOVA (GraphPad Prism 7.03).

## Figures and Tables

**Figure 1 antibiotics-10-00946-f001:**
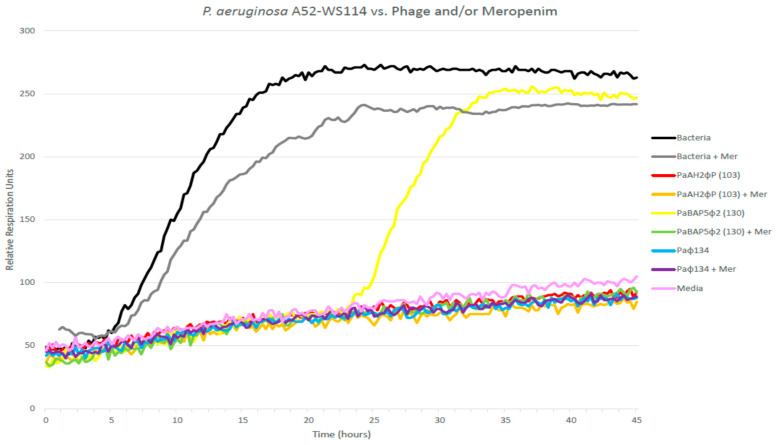
Approximately 10^4^ cfu of *P. aeruginosa* A52-WS 114 were infected separately with phage PaAH2ΦP (103) (red line), PsBAP5Φ2 (130) (yellow line), and PAΦ134 (blue line) in the presence and absence of meropenem (5 µg/mL). The black line portrays bacterial growth in this assay. The gray line (meropenem in isolation) reveals similar bacterial growth in the presence of meropenem confirming resistance. The results indicate that the bacterial isolate was suppressed by each of the three phages independently (red, yellow, and blue profiles). Phage PsBAP5Φ2 (130) inhibited the growth of bacteria up to 22 h before phage-resistant bacteria start dominating the culture (yellow line), while the other two phages could independently suppress bacterial growth out to 45 h. The combination of phage PsBAP5Φ2 (130) with meropenem could suppress bacterial growth out to 45 h, revealing additive effects of this phage–antibiotic combination in targeting (killing) the bacteria.

**Figure 2 antibiotics-10-00946-f002:**
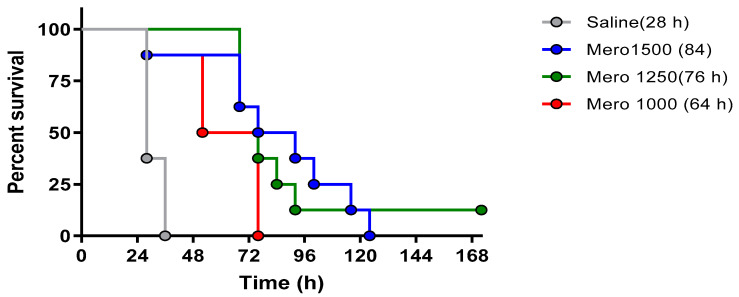
Survival of *P. aeruginosa* infected animals treated with meropenem. BALB/cJ-Cy mice (n = 8 per group) were inoculated with 10^5.5^ cfu of UNC-D by IMIT. Groups received treatments q8 for 120 h of meropenem at doses of 1500 mg/kg/day (blue line), 1250 mg/kg/day (green line), 1000 mg/kg/day (red line), or saline (gray line) initiated 3 h post-bacterial inoculation. Survival was monitored for 168 h. The mean time to death for each group is shown in parenthesis.

**Figure 3 antibiotics-10-00946-f003:**
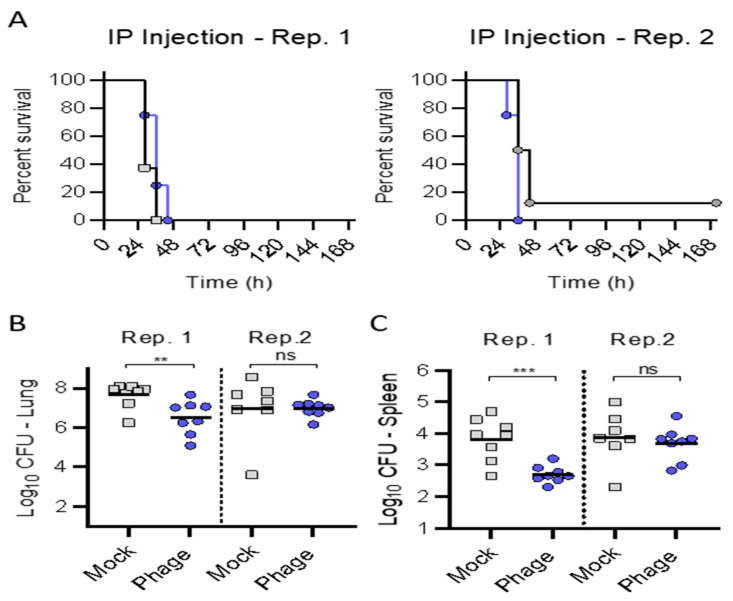
Intraperitoneal administration of phage cocktail against acute pulmonary infection with *P. aeruginosa*. BALB/cJ-Cy mice (*n* = 8 per group) were inoculated with 10^5.5^ cfu of UNC-D by IMIT. Groups received IP injections (q8 for up to 120 h) of phage (1 × 10^9^ pfu; Phage; Gray) or saline (Mock; Blue) 3 h after bacterial inoculation. (**A**) Survival was monitored for 168 h. At the time of euthanasia, bacterial numbers were enumerated in the (**B**) lungs or (**C**) spleen. Data from two independent replicate experiments are are shown (Rep. 1 and Rep. 2). Unpaired student *t*-test: ** *p* < 0.01, *** *p* < 0.001, ns = not significant.

**Figure 4 antibiotics-10-00946-f004:**
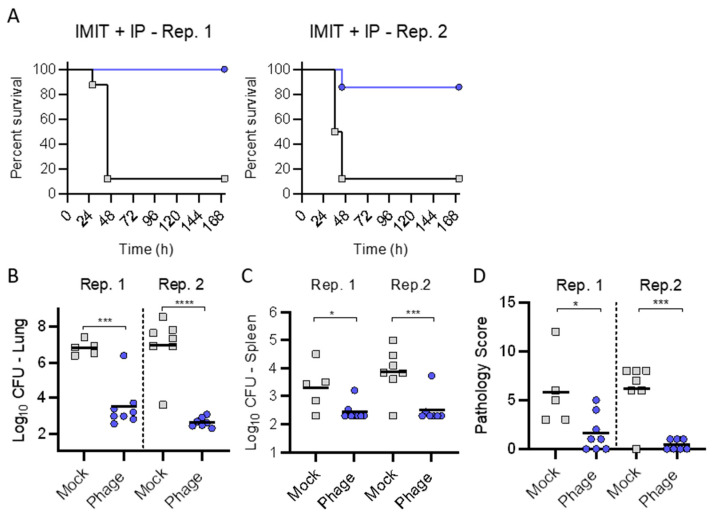
Combined intratracheal and intraperitoneal administration of phage cocktail against acute pulmonary infection with *P. aeruginosa*. BALB/cJ-Cy mice (*n* = 8 per group) were inoculated with 10^5.5^ cfu of UNC-D by IMIT. Groups received a single IMIT administration of phage (2.5 × 10^9^ pfu) 3 h after bacterial inoculation, followed by IP administration of phage (1 × 10^9^ pfu) q8 for 120 h (Phage; Gray) or saline (Mock; Blue). (**A**) Survival was monitored for 168 h. At the time of euthanasia, or at the end of the experiment, bacterial numbers were enumerated in the (**B**) lungs or (**C**) spleen, and (**D**) pathology was scored in the lungs. Data from two independent replicate experiments are are shown (Rep. 1 and Rep. 2). Unpaired student *t*-test; * *p* < 0.05; *** *p* < 0.001; **** *p* < 0.0001.

**Figure 5 antibiotics-10-00946-f005:**
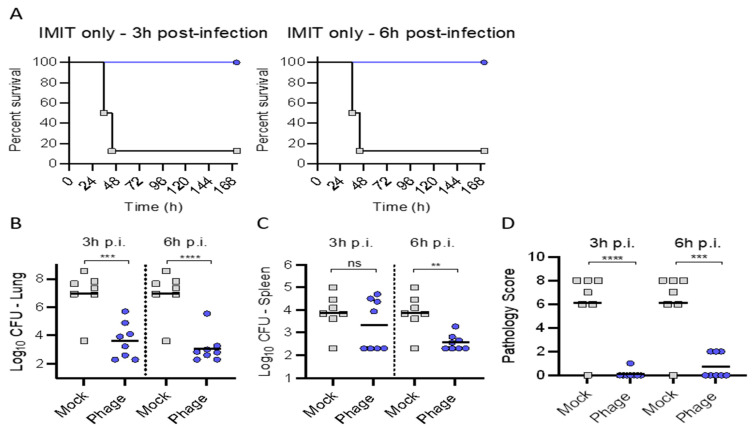
Intratracheal administration of phage cocktail against acute pulmonary infection with *P. aeruginosa*. BALB/cJ-Cy mice (n = 8 per group) were inoculated with 10^5.5^ cfu of UNC-D by IMIT. Groups received a single IMIT administration of phage (2.5 × 10^9^ pfu) 3 or 6 h after bacterial inoculation or saline (Mock; Blue). (**A**) Survival was monitored for 168 h. At the time of euthanasia, or at the end of the experiment, bacterial numbers were enumerated in the (**B**) lungs or (**C**) spleen, and (**D**) pathology was scored in the lungs. Unpaired student *t*-test; ** *p* < 0.01; *** *p* < 0.001; **** *p* < 0.0001; ns = not significant.

**Figure 6 antibiotics-10-00946-f006:**
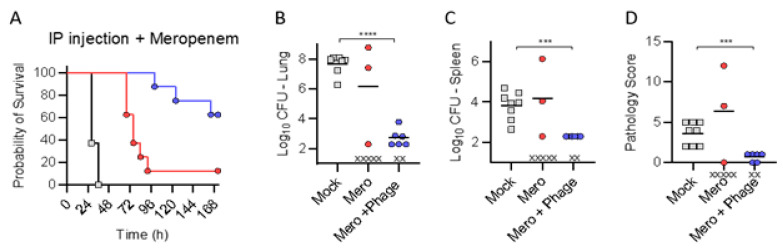
Intraperitoneal administration of phage cocktail during meropenem administration. BALB/cJ-Cy mice (n = 8 per group) were inoculated with 10^5.5^ cfu of UNC-D by IMIT. Beginning 3 h after bacterial inoculation, groups received SQ injections of meropenem (1250 mg/kg/day, q8 for 120 h) and IP injections of phage (1 × 10^9^ pfu; Mero + Phage; Blue) or saline (Mero; Red). Mock treated animals received SQ injections of saline only (Gray). (**A**). Survival was monitored for 168 h. At the time of euthanasia, or at the end of the experiment, bacterial numbers were enumerated in the (**B**) lungs or (**C**) spleen, and (**D**) pathology was scored in the lungs. “x” represent animals that succombed to infection between health checks, and tissues were not harvested. Unpaired student *t*-test (*** *p* < 0.001; **** *p* < 0.0001).

## Data Availability

All data analysis resides at the UofL.

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
