# Peer review of "Successful Intratracheal Treatment of Phage and Antibiotic Combination Therapy of a Multi-Drug Resistant Pseudomonas aeruginosa Murine Model"

_antibiotics, 2021, doi:10.3390/antibiotics10080946_

Round 1

Reviewer 1 Report

The submitted manuscript presents work investigating the effects of phage and antibiotic treatments on survival and bacterial burdens in P. aeruginosa lethal murine model of infections. The language is good, the work is very good and the experimentation has the necessary elements of valid research. Overall, the manuscript illustrates very interesting animal experimentation approaches and describes valuable results but the authors did not do a good job in presenting their work sufficiently clear and understandable to the average reader. The manuscript has several major weaknesses precluding its immediate publication.

  1. Overall, the presentation of the results is unclear and confusing. The facts, units, assays need to clearly stated enabling the readers to follow the work undertaken, the outputs generated and the final findings declared. The title should be changed as even the authors discuss that the synergy could not be proffered from the obtained results.
  2. The abstract could use more clarity, shortening of sentences to make the information precise and understandable to the reader.
  3. The introduction, even though sufficiently referenced does not stress enough the background information leading to the present experimental designs.
  4. Figure 1 and 6 are not legible due to some formatting issue (right part is missing) and the captions are written in a confusing manner so the results are therefore not clear.
  5. Figure 2 is not described nor referenced in the text.
  6. Figure 4 and 6 captions are not sufficiently clear, consider revising (adding title, figure designations e.g. A and B).
  7. In general, the terminology used is not clear nor standardized throughout the text (8TID, 8q, 8qh, IP phage…).
  8. ® should be avoided in scientific literature so this symbol needs to be removed.
  9. The referencing in methods section needs to be adjusted to standard recommended journal reference citations.
  10. Some of the information in methods section (especially in Antibiotic and phage preparations and Antibiotic-Phage combination Assay) is more suitable to results section.
  11. It is not clear why some details of the methodology applied are not specified in methods section but are listed as the Appendices.
  12. Methods section should, in general, be written in the passive voice and "Data not shown" should be avoided.
  13. The use of International System of Units for expressing the results and methodology procedures is not uniform throughout the manuscript, please correct.
  14. The discussion section, in general, covers topics relevant for the obtained results but could benefit from more elaborate argumentation and comparisons with previous studies and how this work adds to the existing knowledge.

Author Response

Reviewer 1

Comments and Suggestions for Authors

The submitted manuscript presents work investigating the effects of phage and antibiotic treatments on survival and bacterial burdens in P. aeruginosa lethal murine model of infections. The language is good, the work is very good and the experimentation has the necessary elements of valid research. Overall, the manuscript illustrates very interesting animal experimentation approaches and describes valuable results but the authors did not do a good job in presenting their work sufficiently clear and understandable to the average reader. The manuscript has several major weaknesses precluding its immediate publication.

  1. Overall, the presentation of the results is unclear and confusing. The facts, units, assays need to clearly stated enabling the readers to follow the work undertaken, the outputs generated and the final findings declared. The title should be changed as even the authors discuss that the synergy could not be proffered from the obtained results.

Response: Changed Title (removing reference to Synergy)

Successful Intra-tracheal treatment of Phage and Antibiotic combination therapy of an Multi-Drug Resistant Pseudomonas aeruginosa Murine Model

  1. The abstract could use more clarity, shortening of sentences to make the information precise and understandable to the reader.

Response:  Edited the abstract as suggested

  1. The introduction, even though sufficiently referenced does not stress enough the background information leading to the present experimental designs.

Response:  Provided more detail in the background.

At the time of our investigation, phage therapy in both pre-clinical and clinical applications for treatment of respiratory infections have been heterogeneous in design. Specifically, we note heterogeneity in phage administration regarding the route (IV, intra-tracheal, both), frequency of administration, and duration (1-3, 5, 13-15).  We therefore wished to systematically evaluate the efficacy of phage therapy in a lethal murine PsA pneumonia model via IV, intra-tracheal, and their combinatorial routes of administration. Additionally we assessed the efficacy of localized instillation of phage for the first time via a one-time administration of IMIT (intubation mediated intra-tracheal) delivery. Finally, we delayed the IMIT mediated phage delivery relative to bacterial inoculation by 6-hours, the longest delay to our knowledge in any PsA pneumonia model to anticipate any potential survivability.  The overarching objectives of this effort was to assess the therapeutic efficacy of phage with and without meropenem  in a lethal cyclophosphamide (Cy) immunocompromised mouse model of MDR P. aeruginosa respiratory disease (pneumonia). Specifically, we evaluated the therapeutic efficacy (survivability, and time to mortality) of phage treatment administered via intubation-mediated intra-tracheal (IMIT) instillation, intraperitoneal (IP) injection, and their combinatorial routes, and as adjunctive treatment with meropenem, in a model achieving persistently high lethality (given resistance to  meropenem). Within all studies undertaken, we evaluated the temperature profiles during the disease course, bacterial burden in the lungs and spleen, and the lung pathology via standard histopathological evaluation.

  1. Figure 1 and 6 are not legible due to some formatting issue (right part is missing) and the captions are written in a confusing manner so the results are therefore not clear.

Response

a). Figure 1 is in landscape

Figure 1. Approximately 104 cfu P. Aeruginosa A52-WS 114 was infected separately with phage PaAH2ΦP(103), PsBAP5Φ2(130) and PAΦ134 in presence and absence of meropenem (5 µg/ml). The black line portrays unadulterated bacterial growth in this assay. The gray line reveals similar bacterial growth in the presence of meropenem confirming resistance. The results indicated that the bacterial isolate was suppressed by each of the 3 phages independently (red, yellow, blue profiles). Phage PsBAP5Φ2(130) inhibited the growth of bacteria up to 22 hours before phage resistant bacteria start dominating the culture (yellow line), while the other two phages could independently suppress bacterial growth out to 45 hours. The combination of phage PsBAP5Φ2(130) with meropenem could suppress bacterial growth out to 45 hours revealing additive effects of this phage antibiotic combination in targeting (killing) the bacteria.  

  1. b) Figure 6

Landscape: Figure edited with Title and Figure Captions

Figure 6. Intraperitoneal administration of phage cocktail protects against pulmonary infection with P. aeruginosa when acute progression is slowed by a sub-efficacious dose of meropenem. BALB/cJ-Cy mice (n=8 per group) were inoculated with 105.5  CFU of UNC-D by IMIT. Groups received treatments of saline placebo (black line), meropenem (1250 mg/kg/day q8 x 5 days) (red line), or the combination of IP phage (1 × 109 pfu) and meropenem  at a dose of 1250 mg/kg/day, both administered q8h x 5 days all treatments initiated 3-hours post bacterial challenge (Figure 6A). Survival was monitored for 168 hours. The combinatorial treatment of meropenem and phage significantly enhanced therapeutic protection (achieving a survival benefit, >50% survival) against pulmonary infection p = 0.0022 (log rank test). We observed significant reductions in the bacterial burden in both the lungs (Figure 6B) and spleen (Figure 6C), accompanied by reduced pulmonary pathologic scores in the combination group relative to saline placebo (“Mock”) (Figure 6D): Unpaired Student t-test (***p<0.001; ****p<0.0001).

  1. Figure 2 is not described nor referenced in the text.

Response: We appended the language in the text and referenced Figure 2 as per below

Survival of P. aeruginosa infected animals treated with meropenem. BALB/cJ-Cy mice (n=8 per group) were inoculated with 105.5  CFU of UNC-D by IMIT. Groups received treatments q8 for 5 days of meropenem at doses of (A) 1500mg (blue line)  (B) 1250mg (green line), or (C) 1000 mg/kg/day (red line) or (D) saline (gray line) initiated 3 hours post bacterial inoculation. Survival was monitored for 168 hours. Mean time to death for each group is shown in parenthesis.

  1. Figure 4 and 6 captions are not sufficiently clear, consider revising (adding title, figure designations e.g. A and B).

Response

a). Figure 6: Edited figure with captions per above

Figure 6. Intraperitoneal administration of phage cocktail protects against pulmonary infection with P. aeruginosa when acute progression is slowed by a sub-efficacious dose of meropenem. BALB/cJ-Cy mice (n=8 per group) were inoculated with 105.5  CFU of UNC-D by IMIT. Groups received treatments of saline placebo (black line), meropenem (1250 mg/kg/day q8 x 5 days) (red line), or the combination of IP phage (1 × 109 pfu) and meropenem  at a dose of 1250 mg/kg/day, both administered q8h x 5 days all treatments initiated 3-hours post bacterial challenge (Figure 6A). Survival was monitored for 168 hours. The combinatorial treatment of meropenem and phage significantly enhanced therapeutic protection (achieving a survival benefit, >50% survivial) against pulmonary infection p = 0.0022 (log rank test). We observed significant reductions in the bacterial burden in both the lungs (Figure 6B) and spleen (Figure 6C), accompanied by reduced pulmonary pathologic scores in the combination group relative to saline placebo (“Mock”) (Figure 6D): Unpaired Student t-test (***p<0.001; ****p<0.0001).

b). Figure 4: Edited figure with title and captions as suggested

Figure 4. BALB/cJ-Cy mice (n=8 per group) were inoculated with 105.5 CFU of UNC-D by IMIT. Survival was monitored for 168-hours.  Combining phage administered via a onetime IMIT administration (2.5 x 109 pfu), followed by IP administration (1 × 109 pfu) q8 x 5 days, we observe a significant survival benefit (only one mouse  died) in either study evaluating phage therapy delayed by 3 or 6-hours (Figures 4A and 4B). Rep1 portrays therapy delayed by 6 hours, while Rep2 represents treatment delay by 3-hours. “Phage” in each graph represents IMIT + IP administered phage, and mock represents the saline controls. Bacterial burdens were significantly reduced in the lungs and spleen with a concomitant reduced pathology score in the lungs (Figures 4C-4E) (Unpaired Student t-test; *p<0.05; ***p < 0.001; ****p<0.0001).

  1. In general, the terminology used is not clear nor standardized throughout the text (8TID, 8q, 8qh, IP phage…).

Response: Standardized Terminology

  1. ® should be avoided in scientific literature so this symbol needs to be removed.

Response: Removed

  1. The referencing in methods section needs to be adjusted to standard recommended journal reference citations.

Response: Converted to standard referencing

  1. Some of the information in methods section (especially in Antibiotic and phage preparations and Antibiotic-Phage combination Assay) is more suitable to results section.

Response: Moved these sections to the results section

  1. It is not clear why some details of the methodology applied are not specified in methods section but are listed as the Appendices.

Response: Moved the balance of the Appendix to the Methods

  1. Methods section should, in general, be written in the passive voice and "Data not shown" should be avoided.

Response:   Removed the “data not shown” and edited section to be in passive voice

  1. The use of International System of Units for expressing the results and methodology procedures is not uniform throughout the manuscript, please correct.

Response: Standardized units

  1. The discussion section, in general, covers topics relevant for the obtained results but could benefit from more elaborate argumentation and comparisons with previous studies and how this work adds to the existing knowledge.

Response: We verified an updated literature search and identified germane pre-clinical and clinical reports regarding PsA mediated PNA and treatment with Phage. As suggested we elaborated upon the rationale for the study design and the results acquired referencing germane literature.  We appended more detail as to how our results add to the current literature. Finally added germane literature for Acinetobacter baumannii mediated pneumonia relevant to our discussion. Specifically appended the following germane articles in the discussion and appended to the reference list

  1. Forti F, Roach DR, Cafora M, Pasini ME, Horner DS, Fiscarelli EV, Rossitto M, Cariani L, Briani F, Debarbieux L, Ghisotti D. Design of a broad-range bacteriophage cocktail that reduces Pseudomonas aeruginosa biofilms and treats acute infections in two animal models. Antimicrob Agents Chemother 2018 62:e02573-17
  2. Wu N., Guo M., Li J., et al. Pre-optimized phage therapy on secondary Acinetobacter baumanniiinfection in four critical COVID-19 patients Emerg Microbes Infect 2021;10(1):612-618
  3. Tan X, Chen H, Zhang M, Zhao Y, Jiang Y, Liu X, Huang W and Ma Y Clinical Experience of Personalized Phage Therapy Against Carbapenem-Resistant Acinetobacter baumannii Lung Infection in a Patient With Chronic Obstructive Pulmonary Disease. Front. Cell. Infect. Microbiol. 2021;11:631585.
  4. Jeon J., Park J., Yong D. Efficacy of bacteriophage treatment against carbapenem-resistant Acinetobacter baumannii in Galleria mellonella larvae and a mouse model of acute pneumonia BMC Microbiol 2019;19:70 s12866-019-1443-5

Reviewer 2 Report

In this current form it is difficult to review this article. For example in Figure1, which colored line represents what kind of phages the authors used or whether meropenem was present or absent was not described in the figure legend. Similar problems can be found in Figure 3-6. In addition, figure legends should be a brief description of experiments the authors performed rather than a place of assessment or discussion about the results. With properly addressing these issues by the authors the reviewer would be able to review this manuscript accordingly.

Author Response

Reviewer 2

Comments and Suggestions for Authors

In this current form it is difficult to review this article.

For example in Figure1, which colored line represents what kind of phages the authors used or whether meropenem was present or absent was not described in the figure legend.

              Response: Edited the legend to provide clarity

 Similar problems can be found in Figure 3-6. In addition, figure legends should be a brief description of experiments the authors performed rather than a place of assessment or discussion about the results. With properly addressing these issues by the authors the reviewer would be able to review this manuscript accordingly.

Response: Edited the legends. Removed aspects of discussion and interpretation and ensured focus on the experiments

Reviewer 3 Report

Dear authors,

Firstly, the manuscript should be prepared in accordance to journal guidelines. there is no running title, background methods and results in the abstract. 

introduction and results are ok, but discussion should be rewritten - at this moment present mainly your results. also, there is some methodology after  APPEDNIX A.

Some information presented is not relevant () and other, basic to the scientific writing, is missing (like stating the main results and contributions of the study to the field in the Discussion).

Also the number of citations is minimal authors should refer to other studies related to phage antibiotics based therapy some of them: doi.org/10.1038/s41522-021-00208-5;  10.1089/mdr.2020.0083, doi.org/10.1111/j.1574-695X.2012.00977.x; 10.3390/ijms21124390 and others... discussion with other will provide additional flow to work.

with respect to Your figures - they should be better organised - most of them go beyond the area of paper and are illegitable in .pdf version.

Should the paper be re-written, organised and corrected, it could be a nice contribution to the phage therapy and combined therapy literature.

Author Response

Reviewer 3

Comments and Suggestions for Authors

Dear authors,

Firstly, the manuscript should be prepared in accordance to journal guidelines. there is no running title, background methods and results in the abstract. 

Response: Edited to conform to guidelines

Introduction and results are ok, but discussion should be rewritten - at this moment present mainly your results. Also, there is some methodology after APPEDNIX A.

Response: Moved the methodologic portions as appropriate to Methods. Eliminated the Appendix

Some information presented is not relevant () and other, basic to the scientific writing, is missing (like stating the main results and contributions of the study to the field in the Discussion).

Response: We edited the discussion to emphasize the main results and contributions of this study

 Our results reveal that a one-time administration of phage via IMIT delivery was 100% efficacious in preventing mortality in an aggressive lethal murine model. IP phage administration coupled to IMIT administration may provide clinical improvement when assessing the bacterial burdens and pulmonary pathologic scores. Finally, IP phage administration was additive to antibiotic therapy in preventing mortality while reducing bacterial burdens and pulmonary pathologic scores confirming a viable clinical therapeutic for PsA mediated pneumonia. This investigation confirms the utility of our aggressive murine model, the utility of IMIT as a reliable precise delivery method for local intra-tracheal phage delivery, and the clinical applicability of combining phage and antibiotics for treatment of refractory bacterial infections.

Also the number of citations is minimal authors should refer to other studies related to phage antibiotics based therapy some of them:

Response: We executed a literature focusing upon phage therapy to treat PsA mediated lower respiratory tract infections. We believe there was a sufficient volume of research solely dedicated to the utilization of phage in both pre-clinical and clinical trials for PsA “pneumonia”. We believe discussing phage lysins and focusing on alternative pathogens is tangential to the discussion section. We appended references relevant to the phage treatment of pneumonia including references related to treatment of Acinetobacter baumanii which we believe is more appropriate to our investigation. Herein, we emphasize the universal efficacy of intratracheal phage therapy to not only PsA but AB pathogens underscoring the applicability of phage therapy to MDR pathogens.

  1. Forti F, Roach DR, Cafora M, Pasini ME, Horner DS, Fiscarelli EV, Rossitto M, Cariani L, Briani F, Debarbieux L, Ghisotti D. Design of a broad-range bacteriophage cocktail that reduces Pseudomonas aeruginosa biofilms and treats acute infections in two animal models. Antimicrob Agents Chemother 2018 62:e02573-17
  2. Wu N., Guo M., Li J., et al. Pre-optimized phage therapy on secondary Acinetobacter baumanniiinfection in four critical COVID-19 patients Emerg Microbes Infect 2021;10(1):612-618
  3. Tan X, Chen H, Zhang M, Zhao Y, Jiang Y, Liu X, Huang W and Ma Y Clinical Experience of Personalized Phage Therapy Against Carbapenem-Resistant Acinetobacter baumannii Lung Infection in a Patient With Chronic Obstructive Pulmonary Disease. Front. Cell. Infect. Microbiol. 2021;11:631585.
  4. Jeon J., Park J., Yong D. Efficacy of bacteriophage treatment against carbapenem-resistant Acinetobacter baumannii in Galleria mellonella larvae and a mouse model of acute pneumonia BMC Microbiol 2019;19:70 s12866-019-1443-5

The suggested references (described below) don’t involve PsA, Pneumonia and phage therapy. There is a large volume of research articles that may be referenced but we feel it best to focus attention on the research involving phage therapy in pneumonia with an emphasis on PsA. As above, we appended the current literature identified in pre-clinical and clinical trials relevant to Acinetobacter pneumonia and phage therapy.

We specifically provide the article titles suggested by the reviewer below. We review the abstracts and believe they are not focused on aspects of pneumonia treatment with phage, nor focused upon phage therapy specific to PsA. 

  • org/10.1038/s41522-021-00208-5; 

NPJ Biofilms Microbiomes

2021 Apr 22;7(1):39.

 doi: 10.1038/s41522-021-00208-5.

Synergistic action of phage phiIPLA-RODI and lytic protein CHAPSH3b: a combination strategy to target Staphylococcus aureus biofilms

Ana Catarina Duarte 1 2, Lucía Fernández 3 4, Vincent De Maesschalck 5 6, Diana Gutiérrez 5, Ana Belén Campelo 1, Yves Briers 5, Rob Lavigne 6, Ana Rodríguez 1 2, Pilar García 1 2

  • 1089/mdr.2020.0083,

2021 Jan;27(1):25-35.

 doi: 10.1089/mdr.2020.0083. Epub 2020 Jun 15.

Environmental Phage-Based Cocktail and Antibiotic Combination Effects on Acinetobacter baumannii Biofilm in a Human Urine Model

Bartłomiej Grygorcewicz 1, Bartosz Wojciuk 2, Marta Roszak 1, Natalia Łubowska 3, Piotr Błażejczak 1, Joanna Jursa-Kulesza 4, Rafał Rakoczy 5, Helena Masiuk 4, Barbara Dołęgowska 1

  • org/10.1111/j.1574-695X.2012.00977.x;

FEMS Immunol Med Microbiol

2012 Jul;65(2):395-8.

 doi: 10.1111/j.1574-695X.2012.00977.x. Epub 2012 May 18.

Synergistic phage-antibiotic combinations for the control of Escherichia coli biofilms in vitro

Elizabeth M Ryan 1, Mahmoud Y Alkawareek, Ryan F Donnelly, Brendan F Gilmore

Affiliations expand

DOI: 10.1111/j.1574-695X.2012.00977.x

(4). 10.3390/ijms21124390 and others... discussion with other will provide additional flow to work.

Int J Mol Sci

 2020 Jun 19;21(12):4390.

 doi: 10.3390/ijms21124390.

Antibiotics Act with vB_AbaP_AGC01 Phage against Acinetobacter baumannii in Human Heat-Inactivated Plasma Blood and Galleria mellonella Models

Bartłomiej Grygorcewicz 1, Marta Roszak 1, Piotr Golec 2, Daria Śleboda-Taront 1, Natalia Łubowska 3, Martyna Górska 4, Joanna Jursa-Kulesza 5, Rafał Rakoczy 6, Bartosz Wojciuk 7, Barbara Dołęgowska 1

Affiliations expand

  • DOI: 3390/ijms21124390

Response: The suggested titles above provide for alternative pathogens,  different clinical syndromes, in vitro biofilm models, phage lysins and focusing on biofilms. We believe this would be extraneous to the focus of our discussion and our specific results which add to the literature regarding the specific personalized phage treatments in PsA mediated PNA models.

Should the paper be re-written, organised and corrected, it could be a nice contribution to the phage therapy and combined therapy literature.

Round 2

Reviewer 1 Report

The quality of the responses is low but the majority of corrections were done and the manuscript is much improved. It can be published after final corrections are made:

  1. Please correct units - use only one q8 or q8h (if the same) and replace degrees F with IS units.
  2. To increase the clarity add abbreviations section with explanation of the non-standard abbreviations used.

Author Response

Reviewer 1

Comments and Suggestions for Authors

The quality of the responses is low but the majority of corrections were done and the manuscript is much improved. It can be published after final corrections are made:

Thanks for the thorough review:

  1. Please correct units - use only one q8 or q8h (if the same) and replace degrees F with IS units.

Done

  1. To increase the clarity add abbreviations section with explanation of the non-standard abbreviations used.

Done (appended a table of abbreviations)

Reviewer 2 Report

Formatting of the manuscript has been better revised than the original version. Figure legends are concisely described for each experiment. Even though the idea of phage therapy for P. aeruginosa infection is not new, the observation of the synergistic effect by both phage therapy and antibiotics (meropenem in this case) is interesting, even though the clear mechanism is unknown. I hope the authors could clarify the mechanism in a near future.

Author Response

Reviewer 2

Comments and Suggestions for Authors

Formatting of the manuscript has been better revised than the original version. Figure legends are concisely described for each experiment. Even though the idea of phage therapy for P. aeruginosa infection is not new, the observation of the synergistic effect by both phage therapy and antibiotics (meropenem in this case) is interesting, even though the clear mechanism is unknown. I hope the authors could clarify the mechanism in a near future.

Thanks for the review

Indeed, we have a grant submission currently under review to focus on phage antibiotic synergy (optimization)

Reviewer 3 Report

N.a.

Author Response

Reviewer 3

Thanks for the review

Comments and Suggestions for Authors

N.a.